# Chemical Profile and Antioxidant and Antimicrobial Activity of *Rosa canina* L. Dried Fruit Commercially Available in Serbia

**DOI:** 10.3390/ijms25052518

**Published:** 2024-02-21

**Authors:** Vojkan M. Miljković, Ljubiša Nikolić, Jelena Mrmošanin, Ivana Gajić, Tatjana Mihajilov-Krstev, Jelena Zvezdanović, Milena Miljković

**Affiliations:** 1Faculty of Technology, University of Niš, Bulevar oslobodjenja 124, 16000 Leskovac, Serbia; nljubisa@tf.ni.ac.rs (L.N.); ivana@tf.ni.ac.rs (I.G.); jzvezdanovic@tf.ni.ac.rs (J.Z.); 2Department of Chemistry, Faculty of Sciences and Mathematics, University of Niš, Višegradska 33, 18000 Niš, Serbia; jelena.mrmosanin@pmf.edu.rs (J.M.); milena.miljkovic@pmf.edu.rs (M.M.); 3Department of Biology and Ecology, Faculty of Sciences and Mathematics, University of Niš, Višegradska 33, 18000 Niš, Serbia; tatjana.mihajilov-krstev@pmf.edu.rs

**Keywords:** *Rosa canina* dried fruit, chemical composition, antioxidant activity, antimicrobial activity

## Abstract

The aim of this work was to give as much information as possible on *Rosa canina* dried fruit that is commercially available in Serbia. In order to provide the chemical composition, the UHPLC-DAD-ESI-MS method was employed for both polar and non-polar extracts of samples obtained with a solvent mixture consisting of hexane, acetone, and ethanol in a volume ratio of 2:1:1, respectively, and 0.05% (*w*/*v*) butylated hydroxytoluene. In addition, the total content levels of lycopene, *β*-carotene, total polyphenols, and flavonoids were determined by means of UV-vis spectrophotometry. The antioxidant activity was tested by applying four different methods: ABTS, DPPH, FRAP, and CUPRAC. Overall, nine compounds were identified. The results of chemical composition analysis were used as the basis for the interpretation of the calculated results for the antioxidant and antimicrobial activity. The obtained results for *R. canina* dried fruit extract are as follows: *β*-carotene—7.25 [mg/100 g fruit weight]; lycopene—2.34 (mg/100 g FW); total polyphenol content (TPC)—2980 [mg GAE/kg FW]; total flavonoid content (TFC)—1454 [mg CE/kg FW]; antioxidant activity—ABTS 12.3 [μmol/100 g FW], DPPH 6.84, FRAP 52.04, and CUPRAC 15,425; and antimicrobial activity—*Staphylococcus aureus* MIC/MMC 4/0 [mg∙mL^−1^], *Enterococcus faecalis* 4/0, *Bacillus cereus* 4/0, *Escherichia coli* 4/0, *Salmonella enteritidis* 4/4, *Enteroabacter aerogenes* 4/0, *Pseudomonas aeruginosa* 2/0, and *Candida albicans* 2/0.

## 1. Introduction

*Rosa canina* L. belongs to the Rosaceae family, which contains over 100 species widely distributed in Europe, Asia, Africa, the Middle East, and North America [1]. This thorny shrub, also known as rose hip, dog rose, or briar rose, is mightily resistant to hard weather conditions [2]. Its pseudo-fruits [3], rose hips, ripen in August–September [4] and gain a brick-red to deep-red color [5]. Rose hips are a valuable resource from both medical and economic perspectives [2].

For the preparation of an aqueous extract (cup of tea), hot water should be poured over 2–5 g of the dried plant material; then, it can be consumed as three to four cups per day as a traditional medicine treatment [3] for a common cold. In addition, rose hips are well known for their efficacy in strengthening the body’s defense against infection [2]. Rose hips are known as the most effective remedy against hemorrhoids and for the treatment of diabetes mellitus [6]. From in vitro, in vivo, and human pharmacological studies, the antioxidative, anti-inflammatory (rheumatoid arthritis and chronic low-back pain), cardioprotective, antiulcerogenic, probiotic, antimicrobial, antimutagenic, and anticancerogenic effects are confirmed [3,7].

Colored rose fruits are good sources of phenolic compounds including tannins, flavonoids, phenolic acids, anthocyanins, and dihydrochalcones [7]. Flavonoids and phenolic acids are secondary plant metabolites and important natural bioactive compounds. The beneficial health effects of *Rosa canina* are partially a result of phenolic activity. It is confirmed that flavonoids such as quercetin, naringenin, rutin, and chlorogenic acid can help against inflammation, cardiovascular diseases, and in cancer prevention [8]. Furthermore, carotenoids (such as lycopene and *β*-carotene) contribute to mediating biochemical reactions in eliminating the negative effects of free radicals. Lycopene can be found in red-colored fruits. It is a strong antioxidant and antimicrobial agent able to prevent degenerative diseases, such as cancer of the lungs, bladder, cervix, prostate, breast, and skin; atherosclerosis; and associated coronary artery disease. *β*-carotene is also an antioxidant [9]. Rose hips are reported to be the richest source of the L-isomer of vitamin C among fruits and vegetables, with a content of 300 to 4000 mg/100 g [7]. Among the plant species that contain the most antioxidants, rose hips have shown the strongest antioxidant properties [10].

The need for healthy food for modern consumers is increasing and, therefore, it is important to find and analyze new products. Rose hips can be consumed fresh and dried. Vitamin C is synthesized only by plants using *L*-galactose or galacturonic acid as precursors, which is an important reason why they should be present in human nutrition [11]. Interestingly, tea made from dried fruits contains more of this vitamin and folate than that from fresh ones [3]. Such fruits can also be processed into jam [12]. The newest utilization of *R. canina* fruit extract is as a natural antioxidant for mayonnaise [13].

Since bioactive compounds are constituents of foods, several techniques have been developed for their separation and identification: mass spectrometry (MS), electrospray ionization mass spectrometry (ESI-MS), liquid chromatography coupled with mass spectrometry (LC/MS), high-performance thin-layer chromatography (HPTLC), nuclear magnetic resonance (NMR), and HPLC-ESI-MS [14]. Electrospray ionization (ESI) in the positive or negative ion mode is the most adequate technique for chemical analysis. Through the positive or negative ion mode, it provides information on the molecular masses [M + H]^+^ or [M − H]^−^ ions and the unconjugated polyphenols, as afterwards, in-source fragmentation products can be identified by comparison with the spectra of authentic standards [10]. For the qualitative and quantitative analysis of phenolics, HPLC-UV/VIS and Folin–Ciocalteu reagent are commonly used by harnessing the reduction capacity of the components from an extract sample. For the determination of antioxidant capacity, in vitro methods such as ABTS, CUPRAC, DPPH, and FRAP are applied [15].

The aim of this study is to examine the chemical composition, as well as the antioxidant and antimicrobial activity, of dried *R. canina* fruit commercially available on the Serbian market and interpret the results from the aspect of chemical composition.

## 2. Results and Discussion

### 2.1. The Identification of Compounds by Chromatographic Analysis

The relationship between a balanced diet and human health is real, and therefore, it is important to identify bioactive compounds in food because providing this information is important for potential customers that pay more attention to the composition of food products. A UHPLC chromatogram for the polar extract with MS detection in the negative mode is given in Figure 1.

A typical UHPLC-DAD chromatogram was recorded using a 300 nm DAD signal (Figure 2). The full list of detected and identified compounds in the extract of dried rose hip is given in Table 1. From the class of phenolic acids and their derivatives, the following compounds were present: caffeic acid, chlorogenic acid, homovanillic acid hexoside, 4-*O*-caffeoyl-quinic acid and caffeic acid (unknown derivative). From the other acids, quinic and citric acids were identified.

Quinic acid was identified by additional ions in the spectrum at *m*/*z* 85 and 127 [22]. For identification of citric acid, referent standard solution was used. Due to the presence of an ion with *m*/*z* 179, caffeic acid was identified [23]. Homovanilic acid hexoside was identified by *m*/*z* 343 and MS/MS fragment ions 181 (100%) and 137 [18]. By characteristic base peaks at *m*/*z* 191 and 173, 4-*O*-caffeoyl quinic acid derivatives were identified. Phloretin *C*-diglycoside with *m*/*z* 597 was tentatively present in both samples [18]. The standard solution of rutin was used for its identification. Naringenin was identified by fragment ions at *m*/*z* 271 and *m*/*z* 151.

In the non-polar extract, two compounds were detected by UHPLC analysis and as-signed as *β*-carotene and lycopene.

Fetni et al. also performed identification of chemical compounds of *R. canina* fruit extract. For preparation of extract, they used an ethanol/water (70:30; *v*/*v*) solvent mixture. Mutual compounds identified with our extract samples were citric acid, caffeic acid, and chlorogenic acid [15]. Mihaylova et al. used water as extraction solvent and identified citric acid, *β*-carotene and lycopene [24]. Compared to the results obtained by Hvattum, who used only flash without seeds of *R. canina* fruits and methanol containing 5 vol. % formic acid, mutual compounds were rutin and quercetin-rhamnoside [10]. Unexpectedly, there was not a single common phenolic compound with the ones identified in *R. canina* extract obtained with 0.3% HCl (*v*/*v*) [13]. Compared to the latest research for *R. canina* fruit extract chemical composition, which dates from 2023, and HPLC with a C18 column and UV/Vis detector, mutual compounds were caffeic acid, chlorogenic acid, rutin and naringenin. In this study, ethanol (80%) was used as extraction solvent [13]. The difference in chemical composition is related to the extraction procedure, solvent used and instrumental method applied.

### 2.2. Total β-Carotene and Lycopene Contents

The data on total *β*-carotene and lycopene contents are of interest because of the nutritive and therapeutic value of these compounds [25]. The carotenoids *β*-carotene and lycopene, with 11 conjugated double bonds in their structure, are effective antioxidants with the ability to quench singlet oxygen. Moreover, it was found that lycopene is the most efficient biological carotenoid singlet oxygen quencher, twice as efficient as *β*-carotene. The extended polyene structure of these carotenoid pigments makes them valuable antioxidant molecules, but at the same time, sensitive to light and/or heat degradation [26]. *β*-carotene has found application as a dietary supplement due to its health-promoting effects [27]. It is the primary plant-based source of dietary vitamin A. The conversion factor of *β*-carotene into vitamin A is 12. In humans, *β*-carotene conversion into vitamin A decreases as the dietary dose increases [28].

Total *β*-carotene and lycopene contents are given in Table 2. The calculated results favor *R. canina* dried fruit as a food source with a high concentration of lycopene. The relationship between balanced diet and human health is real, and therefore, it is important to identify bioactive compounds in food because providing this knowledge is important for potential customers that pay more attention to the composition of food products. The UV/Vis spectrum of a *β*-carotene-lycopene mixture in hexane extract is shown in Figure 3.

The result for total *β*-carotene content of dried rose hips is within the range of the one obtained by Yildiz and Alpaslan, who were determining content in rose hips and three different marmalade samples made of it [29]. In another study that was performed by Turkben et al., total *β*-carotene and lycopene content in fresh rose hips was found to be in the range of 12.9–35.2 mg/100 g FW [30].

It is of interest to compare *β*-carotene content in *R. canina* dried fruits from this study, 7.25 mg/100 g FW, as it is higher than one for black chokeberry (*Aronia melanocarpa*), 1.67 mg/100 g of deseeded berries. Also, for lycopene, dried fruit of *R. canina* showed to be a richer source of lycopene than black chokeberry (*Aronia melanocarpa*), 2.34/100 g FW to 0.6 mg/100 g of deseeded berries [31]. With the recommended daily intake of 900/700 µg of vitamin A for men/women and a conversion factor of *β*-carotene into vitamin A of 12 [28], an individual should take ~150 g fresh mass of rose hips that are commercially available in Serbia to meet the demand for daily vitamin A intake.

The results for *β*-carotene and lycopene contents are related to agronomic factors [32], geographical location and climatic conditions [33]. To have healthy food, all these factors must be fulfilled. Mayo Clinic recommendations for supplementation with *β*-carotene are 6–15 mg per day for adults and teenagers, while children should take 3–6 mg per day [34]. Recommendations for lycopene given by the European Food Safety Authority panel are 0.5 mg/kg body weight per day, at most 75 mg per day [35].

### 2.3. Total Polyphenols and Total Flavonoids Contents

The TPC and TFC are of interest because, in addition to their functions in plants, phenolic compounds in the everyday diet provide health benefits such as antiallergenic, antiatherogenic, anti-inflammatory, antimicrobial, antioxidant, antithrombotic, cardioprotective, and vasodilatatory effects [36]. The phenols are the major antioxidant components [37]. The polyphenols are even added to some food products as supplements [38]. In other words, analysis of total polyphenols and flavonoids provides a determination of the food’s health-promoting potential. Rose hips are considered to be the richest source of all fruit species [2]. Results obtained in this study are given in Table 2.

For comparison with berries, total polyphenols contents extracted from blackberry, blueberry, red raspberry and strawberry are 4350 mg GAE/kg, 3480 mg GAE/kg, 3030 mg GAE/kg, and 2940 mg GAE/kg, respectively [39]. To meet the demands for recommended daily intake of flavan-3-ols (400–600 mg daily) and reduce risk associated with cardiovascular disease and diabetes, 343.7 g of sample should be taken. One cup of dried *R. canina* fruits (150 g) contains 145.5 mg of flavonoids [40].

This is the first report on the chemical composition of *R. canina* dried fruit from Serbia, as Nadpal et al. analyzed *R. dumalis*, *R. dumetorum* and *R. sempervirens* [41]. Compared to them, *R. canina* has lower TPC and TFC.

Although there are numerous studies on the effect of food processes (exposure to light and increased temperature), it is not appropriate to generalize the outcomes of these processes because increases in TPC and TFC can be a result of better extraction after exposure to light and increased temperature, as well as responses to induced stress. On the other hand, decreases are a result of phenolic compound degradation. Therefore, the most objective strategy is to have an individual approach to each food. In a case of *R. canina* infusions obtained with thermal and non-thermal processes, the highest TPC and TFC were reached with thermal processing [42]. This novel research result supports the recipe for rose hip tea with boiled water mentioned in the Introduction. Meanwhile, consumption of fresh rose hips is more beneficial than of dried ones from the aspect of their antioxidant activity [30].

### 2.4. Antioxidant Activity

The beneficial effects derived from phenolic compounds have been attributed to their antioxidant properties, which can protect against non-communicable diseases such as heart disease and cancer involved in reactive oxygen species (i.e., superoxide anion, hydroxyl radicals and peroxy radicals) [43]. The phenolic compounds and especially flavonoids are of interest because of their confirmed antioxidant properties and, therefore, health benefits [44]. Antioxidant ability means scavenging free radicals and oxidative stress reduction. It can be measured by many different assays that are available. The most commonly used are DPPH and ABTS. For each method, a different chromogenic reagent is used. The ABTS method, introduced in 1993, provides greater possibilities compared to DPPH due to the greater polarity of the environment and the solubility in the solvents used [45]; the FRAP method measures only the hydrophilic antioxidants, while DPPH detects only the ones that are soluble in organic solvents; the CUPRAC method is more advanced since it provides both hydrophilic and lipophilic antioxidants to be tested [46]. The popularity, availability, ease of use and susceptibility to modification led to the appearance of many publications in which, because of differences in the methods used, the results are not comparable [45].

Taking into account all of the abovementioned, extract samples of *R. canina* dried fruit commercially available in Serbia were tested by using four different electron transfer-based methods: ABTS, DPPH, FRAP, and CUPRAC. The obtained results are shown in Table 2.

The results achieved in this experimental work confirmed that the antioxidant activity is related to the solvent used for the method applied. That is inconsistent with the conclusion made by Çelik and his coworkers, who performed the same antioxidant activity measuring methods [47].

The Pearson correlation coefficient (Table 3) demonstrates good agreement between the antioxidant assays and the content of total polyphenols and flavonoids.

Pearson’s correlation measures the linear relationship between two continuous sets of data. This coefficient varies from −1 to 1, where a value of −1 indicates a perfect negative correlation, 0 indicates no correlation, and 1 indicates a perfect positive correlation.

The ABTS and DPPH tests employ ABTS and DPPH radicals, which are reduced in the presence of antioxidants, leading to a color change and enabling the assessment of antioxidant activity. The Pearson coefficient indicates the strongest correlation between these two tests (*p* = 0.9672). On the other hand, FRAP and CUPRAC tests involve the reduction of Fe and Cu ions. Since these two tests share similar mechanisms but differ from the previously mentioned tests, the Pearson coefficient confirms a high consistency in the results obtained from these tests (*p* = 0.9986). All tests (ABTS, DPPH, FRAP, CUPRAC) exhibit a high mutual positive correlation, indicating that all of these tests are sensitive to similar changes in samples. From this, it can be inferred that samples demonstrating high antioxidant activity in one test are likely to show activity in other tests as well. The exceptionally high correlation between total flavonoids and DPPH (0.9996) suggests nearly identical changes in these two variables. Therefore, the DPPH test appears to be a key contributor to monitoring total flavonoids. Total polyphenols and total flavonoids indicate a high positive correlation (0.9935), suggesting that these two variables increase together.

Barros et al. tested the antioxidant activity of strawberry-tree berries, sloes and rose hips by the DPPH method, and the results obtained favored rose hips as having the strongest antioxidant activity among them [36]. The scientists from this research group went further and investigated the antioxidant activity of rose hips in different maturity stages and concluded that unripe ones had the highest antioxidant activity [48]. The antioxidant activity of *R. canina* fruit ethanolic extract was higher than that of ascorbic acid and BHT, but lower than that of quercetin and Trolox [15]. Up to now, the value cited in the latest literature data on rose hip fruit extract antioxidant activity, determined by DPPH and expressed as EC_50_ values, is 89.16 µg/mL [13], which is higher than the 428.84 µg/mL determined by Barros et al. [37]. These results are not comparable with the ones obtained in this study because they are expressed differently, although the same method was used, and our results can be compared with the ones that will be obtained by the authors who decide to express them in the same way. However, differences in antioxidant activity can be explained by differences in geographic location and soil composition—which affect the content of phenolic compounds [49]—and by the abovementioned ripening stage.

The phenolic compounds, including anthocyanins, flavonoids, and ascorbic acid, contribute to the antioxidant activities in fruits, and therefore, fruits with higher phenolic contents generally show stronger antioxidant activities. The bioactive compounds with antioxidant activity that are constituents of dried *R. canina* fruit extract are citric acid [50], caffeic acid [51], rutin, naringenin [52], 4-*O*-caffeoyl-quinic acid [53], lycopene and *β*-carotene [54]. Quercetin and chlorogenic acid are not responsible for antioxidant activity [55]. It is of importance to know that besides its own antioxidant activity [56], quinic acid is a pro-metabolite that induces efficacious levels of nicotinamide and tryptophan as antioxidants in humans and, therefore, contributes to nutritional benefits [57].

### 2.5. Antimicrobial Activity

The application of plant extracts as natural antimicrobial agents in the food industry is a growing trend. So, it is *R. canina* fruit extract that found application as a natural antioxidant for mayonnaise [13]. The results of tests for microbial activity performed in this study for *R. canina* dried fruit extract are shown in Table 4.

The results obtained in this study for *R. canina* dried fruit extract are comparable with those obtained by Montazeri and coworkers, who examined the antimicrobial activity of *R. canina* fruit extracts obtained with different solvents. Their results achieved with methanol extracts are the most similar with ours for extract obtained with 70% (*v*/*v*) ethanol. In more detail, they found the MIC/MBC against *Staphylococcus aureus*, *Bacillus cereus*, *Escherichia coli*, and *Candida albicans* to be [mg∙mL^−1^] 2.0/10.5; 4.5/12.5; 5.0/15.5 and 2.5/5.5 and concluded that antimicrobial activity varies depending on the solvent used [58]. Yilmaz and Ercisli, who used methanol for rose hip extraction and tested the antimicrobial activity of that extract against *Enterococcus faecalis* and *Bacillus cereus*, identified an MIC [mg∙mL^−1^] of 0.5 and 0.0625, respectively [59]. Oyedemi et al., who tested the antimicrobial activity of crude *R. canina* fruit extract, determined MICs against *Pseudomonas aeruginosa* and *Escherichia coli* of 0.256 [mg∙mL^−1^] and >0.512 [mg∙mL^−1^], respectively [60].

For the determination of antimicrobial activity in this experimental work, extract in full was used, as suggested by Liu. That is because the total antimicrobial effect is a result of a synergistic effect of each phytochemical present in the extract [61]. Concretely, quinic acid itself possesses antimicrobial activity against *Staphylococcus aureus* [62]. Citric acid is effective against *Staphylococcus aureus* [55], *Klebsiella aerogenes* [63] and *Escherichia coli* [55,63,64]. It is reported that caffeic acid’s MIC is 0.256 [mg∙mL^−1^] [65] and 0.625 [mg∙mL^−1^] [66] against *Staphylococcus aureus* 6538, the same strain that was used in this test. In addition, it has the ability to augment the effects of antibiotics [65]. The antimicrobial activity of chlorogenic acid against *Staphylococcus aureus* was also reported [67]. Rutin’s MIC against *Staphylococcus aureus*, *Enterococcus faecalis*, *Escherichia coli*, *Pseudomonas aeruginosa* and *Candida albicans* was achieved with concentrations [mg∙mL^−1^] of 16, 8, 16, 16, and 16, respectively [68]. Lycopen’s antimicrobial activity against *Staphylococcus aureus*, *Escherichia coli*, *Pseudomonas aeruginosa* and *Candida albicans* was also experimentally confirmed [69]. Naringenin’s antimicrobial activity was demonstrated against both Gram-positive (*Bacillus cereus* ATCC 27348 and *Staphylococcus aureus* ATCC 12598) and Gram-negative (*Escherichia coli* and *Pseudomonas aeruginosa* ATCC 10145) bacterial strains [70].

## 3. Materials and Methods

### 3.1. Plant Material

Dried *R. canina* fruits were bought in the local supermarket in Niš, Serbia. Declared by the producer, the percentage of dry matter was 92.7%, and the country of origin Turkey. After they were washed with tap water, they were left to dry in the shade. Subsequently, the dried rose hips were then sliced in a Braun^®^ blender (Braun, Solingen, Germany) and homogenized roughly.

### 3.2. Chemicals

Acetone was obtained from Fisher Scientific (Loughborough, UK). Hexane, butylated hydroxytoluene, sodium acetate, ethanol, glacial acetic acid, 2,2-diphenyl picrylhydrazyl (DPPH), and 6-hydroxy-2,5,7,8-tetramethylchroman-2-carboxylic acid (Trolox) were purchased from Sigma-Aldrich (Steinheim, Germany).

### 3.3. Extraction Methods

For lycopene and *β*-carotene determinations, samples were prepared by the method described [71]. Briefly, a known quantity of dried rose hips was added to a mixture consisting of organic solvents (n-hexane ≥ 99%, acetone ≥ 99%, ethanol 96% from Sigma Aldrich, St. Louis, MO, USA in a volume ratio of 2:1:1) and 50 mg∙L^−1^ butylated hydroxyl toluene (Merck, Darmstadt, Germany). These mixtures were stoppered and mixed on an orbital shaker in a water bath at 3 Hz for 15 min at 5 °C. Then, 75 mL of cold deionized water for each 10 g of the starting samples was added to the mixture, which was agitated for another 5 min. The suspension was then transferred to a separation funnel to separate the upper (non-polar) phase from the lower (polar) phase for 10 min at laboratory temperature (25 °C).

The polar extract was obtained by homogenizing a determined amount of the sample and mixing it with three times the amount of ethanol (70% *v*/*v*). Afterwards, this mixture was stirred on a magnetic stirrer for 10 min at 5 °C in a water bath and then centrifuged at 8000× *g* for 10 min. The obtained supernatant was decanted, and the remaining sample was re-extracted by the method described above, with twice the volume of ethanol. The supernatants obtained were put together and made up to a known volume with 70% (*v*/*v*) ethanol.

For the purpose of ultra-high performance liquid chromatography (UHPLC), non-polar extracts were dried in a stream of nitrogen and then dissolved in the same volume of methanol, with filtration by 0.45 µm filter prior to analysis, while polar extracts were only subjected to filtration.

### 3.4. UV-Vis Analysis

The non-polar phase was processed by UV-Vis spectrophotometric analysis. Determination of the lycopene, *β*-carotene, total polyphenols and flavonoids contents along with antioxidant activity was performed by spectrophotometer UV-1800 (Shimadzu, Kyoto, Japan). The lycopene and *β*-carotene concentrations in the samples were determined by measuring the absorbance of the non-polar layer in 1 cm path length glass cuvettes at 450 nm and 503 nm, respectively, versus a blank of the above-mentioned solvent mixture. The contents of these carotenoids were calculated from the system of linear equations:*A*_450_ = ε*_L_*_450_C*_L_* + ε*_β_*_450_C*_β_*(1)
*A*_503_ = ε*_L_*_503_C*_L_* + ε*_β_*_503_C*_β_*(2)
where *A*_450_ and *A*_503_ are absorbances at 450 nm and 503 nm, respectively, ε*_L_*_450_ and ε*_L_*_503_ are molar absorption coefficients of lycopene at 450 nm and 503 nm, respectively, and ε*_β_*_450_ and ε*_β_*_503_ are molar absorption coefficients of *β*-carotene at 450 nm and 503 nm, respectively. Here, C*_L_* and C*_β_* stand for concentrations of lycopene and *β*-carotene, respectively (expressed as moles per liter). This unit refers to the working solution used in the UV-Vis method, while lycopene and *β*-carotene contents were calculated and expressed in milligrams per kilogram.

### 3.5. UHPLC-ESI-MS Analysis

Both the polar and non-polar extracts were processed by qualitative UHPLC analysis. The UHPLC analysis was performed on a Hypersil gold C18 column (50 mm × 2.1 mm, 1.9 μm) at 25 °C using a Dionex Ultimate 3000 UHPLC+ system equipped with a diode array (DAD) detector and LCQ Fleet Ion Trap Mass Spectrometer (Thermo Fisher Scientific, Waltham, MA, USA). The method applied was described by [72]. For the non-polar extract, an isocratic method with methanol and acetonitrile at 1:1 (*v*/*v*) at 0.25 mL∙min^−1^ flow rate was applied. The injection sample volume was 4 μL.

For the polar extract, the flow rate of the mobile phase was set to 0.250 mL∙min^−1^, while the volume of the sample was 8 μL. A 3D-ion trap with electrospray ionization (ESI) in positive ion mode was used for mass spectrometric analysis of the non-polar extract, while for the polar extract, negative and positive ion modes were applied. The mass spectra were acquired by full range acquisition of *m*/*z* 100–700, with a tandem mass spectrometry analysis performed by a data-dependent scan, with the collision-induced dissociation of detected molecular ions peaks ([M − H]^−^/[M + H]^+^) tuned at 30 eV for both ionization modes. For both modes, the capillary temperature was 350 °C, and nitrogen sheath and auxiliary gas flow were 32 and 8 arbitrary units, respectively. For instrument control, data acquisition and data analysis, Xcalibur software version 2.1 (Thermo Fisher Scientific) was used.

The qualitative analysis was based on comparison of retention times and MS spectra with the corresponding molecular ion peaks as well as the characteristic ion fragmentation of selected peaks (MS/MS) from corresponding UHPLC chromatograms, and comparison with a mass spectral database available online (MassBank, MassBank consortium) and the available literature. Full identification was provided by using reference standards for some compounds (citric acid, chlorogenic acid and rutin dihydrate, all from Merck). Methanol, acetone, and water (LC-MS purity) from Thermo Fisher Scientific were used in the mobile phase along with formic acid of HPLC purity obtained from Carlo Erba (Emmendingen, Germany).

### 3.6. Total Polyphenols Content

The TPC of dried *R. canina* fruits was determined by the method with Folin–Ciocalteu reagent described by Huang et al. [18]. In short, 0.4 mL of previously prepared and defatted sample was mixed with 0.5 mL of Folin–Ciocalteu reagent and 2 mL of 20% Na_2_CO_3_ solution in a volumetric flask of 10 mL. The flask was filled with deionized water to the line and incubated for 2 h at 20 °C, and absorbance was measured at 760 nm, relative to deionized water as a reference solution. The calibration line was obtained, and it was linear in the concentration range from 1 mg∙L^−1^ to 9 mg∙L^−1^. Compared to the method performed by Swain and Hillis in 1957, who used Folin–Denis reagent for total phenols content [73], this one is more optimized, which can be seen in more intense blue/green color in a reaction with phenols. Because of lithium sulfate addition into Folin–Ciocalteu reagent, a difference between these two methods is in the reduced precipitation of white salts that form upon reaction with Folin–Denis reagent. 

Calculation of the TPC in the tested samples was performed by the equation of the calibration line and expressed as milligrams of gallic acid equivalents (GAE) per kilogram of the sample. The following equation was used:*A* = 0.10262*c_x_* + 0.05719 (*r*^2^ = 0.999469)(3)
where *A* is absorbance, *c_x_* is analyte concentration and *r*^2^ is the coefficient of determination.

### 3.7. Total Flavonoids Content

The method applied for the determination of TFC was described by de Souza et al. [19]. The reaction mixture was prepared by mixing 0.25 mL of the sample, 3 mL of deionized water and 0.3 mL of 5% NaNO_2_. After 5 min, 1.5 mL of AlCl_3_ was added to this mixture, and then, after another 5 min, 2 mL of 1 mol·L^−1^ NaOH and deionized water was added up to 10 mL. The absorbance of the solution was measured at 510 nm in relation to deionized water as a reference solution by UV-Vis spectrophotometer. The blank solution consisted of all substances but without the real sample. A series of working solutions were prepared from the starting solution of (+)-catechin in order to construct a calibration curve. It was linear in the concentrations range from 5 mg∙L^−1^ to 40 mg∙L^−1^. On the basis of the obtained equation for the calibration line (Equation (4)), the TFC was calculated and expressed as milligrams of catechin equivalents (CE) per kilogram of the sample:*A* = 0.03612*c_x_* + 0.0091 (*r*^2^ = 0.9993)(4)
where *A* is absorbance, *c_x_* is analyte concentration, and *r*^2^ is the coefficient of determination.

### 3.8. Antioxidant Activity by ABTS Method

The 2,2′-azino-bis(3-ethylbenzothiazoline 6-sulfonic acid (ABTS) method [74] is principally based on decolorization of a blue-green ABTS radical cation formed by chemical or enzymatic oxidation of ABTS solution. By this method, 0.1 mL of the dried fruit sample was mixed with 2 mL of ABTS radical cation working solution. After the solution was kept in the dark for 6 min, absorbance was measured at 734 nm relative to methanol as the reference solution. The absorbance values of a series of standard solutions were subtracted from the absorbance of the blank. Graphic dependence is given by Equation (5):Δ*A* =*A*_0_ − *As* = (*c_x_*)(5)
where *A*_0_ is the absorbance of the blank and *A_S_* is the mean value of three samples of the standard solutions, which have known concentrations. The calibration plot was linear in the concentration range from 0.5 µmol∙L^−1^ to 2 µmol∙L^−1^, and it is given by Equation (6):Δ*A* = 0.0316*c_x_* + 0.0068 (*r*^2^ = 0.9998)(6)
where *c_x_* is ABTS radical cation concentration expressed in micromoles per liter. Based on the obtained equation, the antioxidant activity was calculated and expressed as millimoles of Trolox equivalents (TE) per kilogram of the sample.

### 3.9. Antioxidant Activity by DPPH Method

The 2,2-diphenyl-1-picrylhydrazyl (DPPH) method [75] is based on a mechanism similar to that of the previously described ABTS method. A purple solution of DPPH radicals reacts with antioxidants in the tested samples. These radicals are reduced to yellow DPPH form, which is followed by a decrease in absorbance at 515 nm. By this method, 0.1 mL of the sample was mixed with 5 mL of DPPH radical working solution in a volumetric flask and filled up to 10 mL with methanol. After standing at 20 °C for 6 min, the absorbance of the resulting solution was measured at 734 nm, relative to methanol as the reference solution. A series of standard solutions were prepared by adding 5 mL of DPPH to a certain volume of Trolox and filling up to 10 mL with methanol. After 30 min, absorbance was measured and was given as the mean of three measurements. The absorbance values of the series of standard solutions were subtracted from the absorbance of the blank. The calibration plot was linear in the concentrations range from 0.5 µmol∙L^−1^ to 5 µmol∙L^−1^ and had the form:Δ*A* = 0.02449*c_x_* + 0.00913 (*r*^2^ = 0.9988)(7)
where *c_x_* is DPPH radical concentration expressed in micromoles per liter. Based on the obtained equation, the antioxidant activity was calculated and expressed as millimoles of TE per kilogram of the sample.

### 3.10. Antioxidant Activity by FRAP Method

Determination of antioxidant activity by the ferric ion reducing antioxidant power (FRAP) method is based on the formation of *o*-phenanthroline-Fe^2+^ complex and its degradation in the presence of chelating agents [76]. Practically, the test solution (20 μL) was diluted with 0.38 mL of deionized water, and 3 mL of FRAP reagent (mixture of acetate buffer, 2,4,6-tripyridyl-S-triazine (TPTZ) and FeCl_3_ in ratio 10:1:1 (*v*/*v*/*v*) was added. The mixture was incubated for 5 min at 37 °C, and absorbance was measured at 595 nm in relation to the blank, which contained the solvent instead of the sample. A series of standard solutions of FeSO_4_ · 7H_2_O were used to obtain the calibration line, which was linear in the concentration range from 1.39 µmol∙L^−1^ to 13.9 µmol∙L^−1^. Based on the obtained equation of the line, antioxidant (FRAP) activity was calculated and expressed as micromoles of Fe^2+^ equivalent per kilogram of the sample. The calibration line was defined as:Δ*A* = 0.077*c_x_* + 0.0286 (*r*^2^ = 0.9999)(8)
where *c_x_* is concentration of Fe^2+^ ions expressed in millimoles per liter.

### 3.11. Antioxidant Activity by CUPRAC Method 

The cupric ion reducing antioxidant capacity (CUPRAC) method applied for this test was described previously by Apak et al. [77]. It is based on formation of copper(I)-neocuproine complex, which shows maximum absorption at 450 nm. In our work, a series of standard solutions of Trolox were used to obtain the calibration line, which was linear in the concentration range from 3 µmol∙L^−1^ to 18 µmol∙L^−1^. Based on the obtained equation of the line, antioxidant (CUPRAC) activity was calculated and expressed as micromoles of TE per kilogram of the sample. The calibration line was defined as:Δ*A* = 0.0606*c_x_* + 0.0449 (*r*^2^ = 0.9993)(9)
where *c_x_* is concentration of Trolox expressed in millimoles per liter.

### 3.12. Antimicrobial Activity 

The antimicrobial activity testing of the samples was performed against microorganism strains from the laboratory collection. The Gram-positive bacteria used for the tests were *Staphylococcus aureus* (ATCC 6538), *Enterococcus faecalis* (ATCC 19433) and *Bacillus cereus* (ATCC 11778). The Gram-negative bacteria were *Escherichia coli* (ATCC 25922), *Salmonella enteritidis* (ATCC 13076), *Enterobacter aerogenes* (ATCC 13048) and *Pseudomonas aeruginosa* (ATCC 9027). The yeast *Candida albicans* (ATCC 24433) was also used for testing the antimicrobial activity. Overnight culture on oblique Mueller–Hinton agar (Institute of Virology, Vaccines and Sera “Torlak”, Belgrade, Serbia) was prepared for bacterial analysis, while oblique Sabouraud dextrose agar (Institute of Virology, Vaccines and Sera “Torlak”) was used for the yeast.

The microdilution method was applied to test the antimicrobial activity of the extracts. The overnight cultures of selected strains of microorganisms were used to make suspensions of 0.5 McFarland turbidity corresponding to a density of 1 × 10^8^ CFU∙mL^−1^. Two types of sample solutions were prepared from the extract of *R. canina* dried fruit. The first type of extract was obtained by using a mixture of solvents (250 mL of hexane, 125 mL of acetone, 62.5 mL of ethanol, and 62.5 mL of 50 mg∙L^−1^ butylated hydroxytoluene), while dry residues were dissolved in 100% dimethyl sulfoxide (DMSO; Merck). The second type of extract was prepared by using 70% ethanol as solvent. Volumes of 160 μL of inoculated Mueller–Hinton broth were introduced into the microtiter plate with 96 wells, and 40 μL of the initial sample and a series of binary dilutions were pipetted. The final volume in each well was 100 μL with density of microorganisms of 10^6^ CFU∙mL^−1^. Cultivation of all tested microorganisms was carried out at 37 °C for 18 h according to the recommended Clinical and Laboratory Standards Institute procedure [78]. For *Candida albicans*, this testing time was enough because the test was not meant to determine the exact number of cells (viable counts) but only to observe whether culture growth (inhibitory activity of the extract) or cell death (microbicidal activity) occurred.

The minimum inhibitory concentration (MIC) is defined as the concentration of the sample in which there is no visible growth of microorganisms. It was determined by using 5 g∙L^−1^ of aqueous solution of triphenyltetrazolium chloride (Sigma Aldrich). The minimum microbicidal concentration (MMC) is defined as the concentration of sample that is capable of killing 99.9% of microbial cells. It was determined by transferring the contents of wells with no visible growth to Petri dishes with Mueller–Hinton agar for bacteria and Sabouraud dextrose agar for yeasts, incubating them and counting colonies. All tests were performed in triplicate, and the obtained results were processed by analysis of variance (ANOVA) with 95% confidence (*p* ≤ 0.05).

## 4. Conclusions

The UHPLC method showed to be adequate in providing the results for chemical composition, which were the proper basis for antioxidant and antimicrobial activity results interpretation. The bioactive compounds present in dried *R. canina* polar fruit extract are quinic acid, citric acid, caffeic acid, chlorogenic acid, homovanilic acid hexoside, 4-*O*-caffeoyl-quinic acid, phloretin-*C*-diglycoside, rutin, quercetin-hexoside-rhamnoside, naringenin, lycopene and *β*-carotene. The results for antioxidant activity confirmed their dependence on the solvent used. The results for antimicrobial activity obtained in this study expanded the antimicrobial profile of *R. canina* species since not all of the bacterial strains used in this study were used in the studies conducted by other authors. Overall, results obtained in this study recommend *R. Canina* dried fruits that are commercially available in Serbia as a rich source of health-beneficial bioactive compounds. Further studies should be performed in order to test the extract application as a possible food additive that contributes to the extension of food product shelf life.

## Figures and Tables

**Figure 1 ijms-25-02518-f001:**
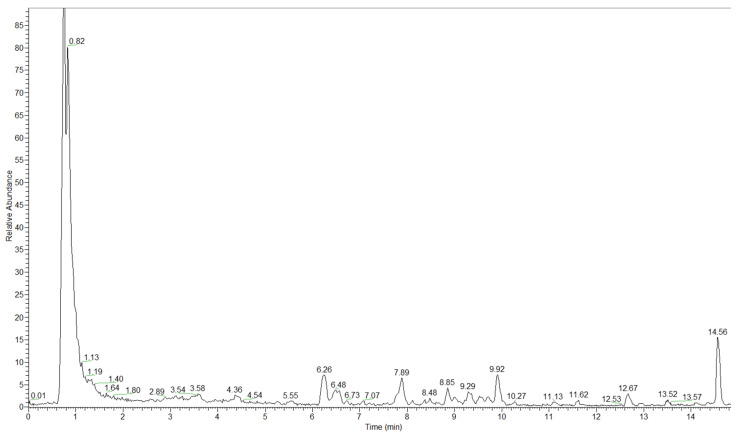
UHPLC chromatogram for the polar extract sample with MS detection in negative mode.

**Figure 2 ijms-25-02518-f002:**
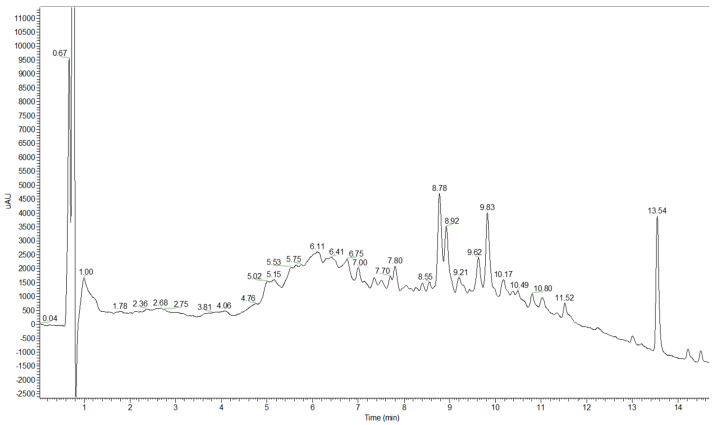
UHPLC-DAD chromatogram for the polar extract.

**Figure 3 ijms-25-02518-f003:**
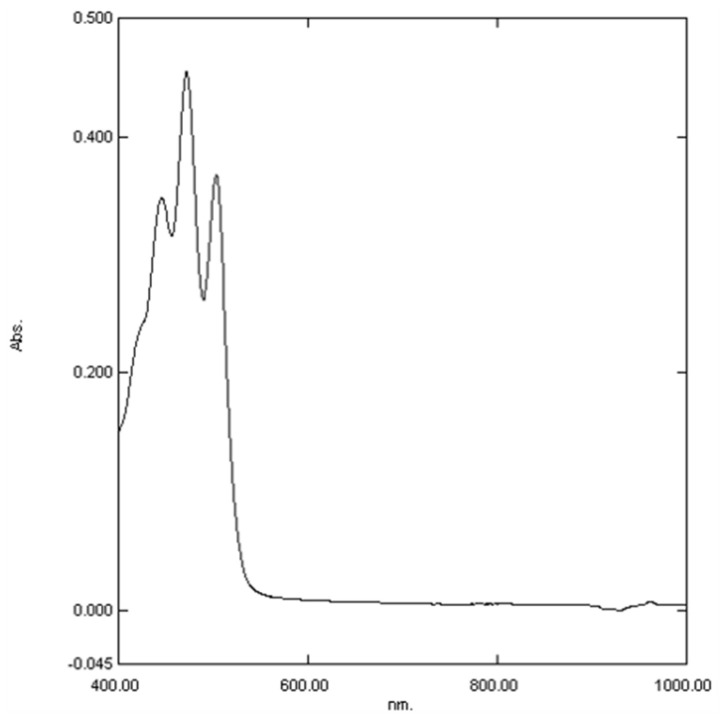
UV/Vis spectrum of a *β*-carotene-lycopene mixture in hexane extract.

**Table 1 ijms-25-02518-t001:** Compounds detected by UHPLC-DAD-MS/MS.

Peak	*t*_R_ [min]	λ_max_ [nm]	Molecular Ion [M − H]^−^ *m*/*z*	MS/MS Fragment Ions *m*/*z*	Assignment	Presence in Sample
1.	0.75	−	191	173, 127, 111, 85 (100%)	Quinic acid [16]	+
2.	0.92	−	191	173, 111 (100%)	Citric acid (standard)	+
3.	1.50	−	179	161, 143, 131, 119, 113, 101, 89 (100%), 71	Caffeic acid [17]	+
4.	5.30	325 303sh	353	191 (100%), 179, 173	Chlorogenic acid (standard)	+
5.	5.40	318 298sh	343	181 (100%), 137	Homovanilic acid hexoside [18]	+
6.	5.50	−	477	431 (100%)	not identified	+
7.	5.76	325 300sh	353	191, 179, 173 (100%), 135	4-*O*-caffeoyl-quinic acid	+
8.	6.23	−	457	411 (100%)	not identified	+
9.	6.63	−	457	411 (100%), 341	not identified	+
10.	8.18	354 262	471	425 (100%), 263	not identified flavonoid	+
11.	8.99	351 289	597	487, 387, 357 (100%)	Phloretin-*C*-diglycoside [19]	+
12.	8.99	355 258	609	301 (100%), 299, 271	Rutin (quercetin-3-*O*-rutinoside) (standard)	+
13.	9.26	321 293	579	533, 459 (100%), 357, 313, 271, 235	not identified	+
14.	9.48	324 295	503	324 295	not identified	+
15.	9.70	−	609	489, 301 (100%)	Quercetin-hexoside-rhamnoside (tent.)	+
16.	10.90	−	537 ^#^	457 (100%)	Lycopene or *β*-carotene [20]	+
17.	11.40	291	271	227, 177, 151 (100%), 107, 93	Naringenin [16,21]	+
18.	13.47	343	386 ^#^	201 (100%)	not identified	+
19.	14.25	358	274 ^#^	256 (100%), 230, 106, 102, 88	not identified	+

^#^ Electrospray Ionization Mass Spectrometry (ESI-MS) data are corresponding to positive mode, ([M + H]^+^), sh—shoulder. Presence in sample: (+)—compound is present, (−)—compound is not present, tent.—tentatively. Values in the brackets in the column of MS/MS fragment ions mean ion abundance of the base ion peak in the corresponding MS/MS spectrum of the compound.

**Table 2 ijms-25-02518-t002:** Total content of *β*-carotene and lycopene, TPC and TFC, antioxidant activity of *R. canina* dried fruit extract.

Total *β*-Carotene and Lycopene Contents [mg/100 g FW]		TPC and TFC [mg·kg^−1^]		ABTS	DPPH	FRAP	CUPRAC
				Csr ± SD [μmol/100 g FW]			
7.25 ± 0.09 (RSD = 7.04%)	2.34 ± 0.06 (RSD = 2.35%)	2980.94 ± 7 (RSD = 2.3%)	1454.73 ± 20 (RSD = 1.35%)	12.30 ± 0.78	6.84 ± 0.03	52.04 ± 1.02	15,425.38 ± 35.86

The TPC is expressed as milligrams of gallic acid equivalents (GAE) per kilogram fresh weight of the sample. The TFC is expressed as milligrams of catechin equivalents (CE) per kilogram fresh weight of the sample. RSD—relative standard deviation. ABTS—antioxidant activity determined by 2,2′-azino-bis(3-ethylbenzothiazoline-6-sulfonic acid) method (expressed as millimoles of Trolox equivalents per kilogram of the sample), DPPH—antioxidant activity determined by 2,2-diphenyl-1-picrylhydrazyl method (expressed as millimoles of Trolox equivalents per kilogram of the sample), FRAP—antioxidant activity determined by ferric reducing antioxidant power method (expressed as micromoles of Fe^2+^ equivalents per kilogram of the sample), CUPRAC—antioxidant activity determined by cupric ion reducing antioxidant capacity method (expressed as millimoles of Trolox equivalents per kilogram of the sample).

**Table 3 ijms-25-02518-t003:** Pearson’s correlation coefficient.

	Total Polyphenols	Total Flavonoids	ABTS	DPPH	FRAP	CUPRAC
Total polyphenols	1.0000					
Total flavonoids	0.9935	1.0000				
ABTS	0.9935	0.9740	1.0000			
DPPH	0.9899	0.9996	0.9672	1.0000		
FRAP	0.9861	0.9986	0.9607	0.9997	1.0000	
CUPRAC	0.9935	0.9982	0.9740	0.9996	0.9986	1.0000

**Table 4 ijms-25-02518-t004:** Antimicrobial activity of *R. canina* dried fruit extract.

Microbial Strain	Minimum Inhibitory (MIC)/ Minimum Microbicidal Concentration (MMC) [mg∙mL^−1^]
Gram (+) bacteria	
*Staphylococcus aureus*	4.0/0
*Enterococcus faecalis*	4.0/0
*Bacillus cereus*	4.0/0
Gram (−) bacteria	
*Escherichia coli*	4.0/0
*Salmonella enteritidis*	4.0/4.0
*Enteroabacter aerogenes*	4.0/0
*Pseudomonas aeruginosa*	2.0/0
Yeast	
*Candida albicans*	2.0/0

## Data Availability

The data used to support the findings of this study are available upon request from the corresponding author.

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
