# Peer review of "Chemical Profile and Antioxidant and Antimicrobial Activity of Rosa canina L. Dried Fruit Commercially Available in Serbia"

_ijms, 2024, doi:10.3390/ijms25052518_

Round 1

Reviewer 1 Report

Comments and Suggestions for Authors

This is a well-organized research project and a well-written manuscript that includes interesting and useful analyses of antimicrobial and antioxidant activities. The manuscript includes 71 references covering important literature sources.

I only suggest some minor corrections that can additionally improve the quality of the work, most of which are typos or technical nature:

-        -  Latin names (species and genus) in italics (e.g. Abstract, line 261), but not for the family name (e.g. Introduction); Latin names of microorganism species should be written correctly, using 1st capital letter (genus name) and 2nd word written in lowercase letters (species name) (e.g. Abstract, lines 267, 273, 279 etc.); all of this should be corrected according to botanical standards.

-       -   Introduction, line 36: passive form of the sentence should be used.

-      -    If using R. canina as an abbreviation (for the genus), it should be used everywhere after the 1st mention of the full name at the very beginning in the Introduction – it is not appropriate use the both forms randomly.

-       -   I suggest revising the title and changing ‘’from supermarket’’ to ‘’commercially available’.

-       -   In the Material and Methods section, it could be much more appropriate to highlight the country of origin of the sample, especially since differences in chemical composition may be the result of origin, and not only ‘’extraction procedure, solvent used and instrumental method applied’’ (line 124 – which I also propose to correct). 

-       -   The conclusions are in accordance with the evidence presented. The antioxidant activity were evaluated by 4 methods (DPPH; FRAP; CUPRAC; ABTS), and antimicrobial activity by 1 method (microdilution method), all of them being the standard procedures used for the analysis of biological activities.

-       -   Chromatograms and spectrums may be attached as a supplement, not the main text.

-       -   The references are appropriate.

Reviewer 2 Report

Comments and Suggestions for Authors

The manuscript entitled “Chemical profile, antioxidant and antimicrobial activity of Rosa canina L. dried fruit from supermarket in Serbia” contains interesting data. The objective is properly stated. Comments addressed to the authors are below:

Introduction is well written.

The section Results and Discussion should be improved. Authors only compared their results with data published by other researchers. Thay could give some explanation of obtained results.

In case of vitamin C and B-carotene content the discussion concerning the effect of various thermal treatment on content of this nutrient should be add. Generally authors should also add information about the possibility of using fruit in human nutrition. Please compare the content vitamin C with the demand for  it in various groups  of population (based on age) in your country and other European countries.

Please give the information about the conversion factor of b-carotene to vitamin A and also provide information about the meeting of demand for vitamin A.

Similar comment is to phenolic compounds and flavonoids. Please provide information about the effect of various processes including drying on the level of phenolic compounds and flavonoids. Please compare the content of both mentioned above groups with the demand in human nutrition.

The results from tables 2, 3, 4 should be put  in one table.

Materials and  Methods section:

The authors stated that they bought dry samples but how usually are samples were dry. Please add these information to the M&M section.

Please clarified how samples were homogenised. The information regarding the type of homogeniser should be add.

The extraction for samples preparation or the UHPLC-ESI-MS  method should be improved. They are described but not cleary.

The by the Folin-Ciocalteu method this is not method this is solvent for the total phenolic content determination. Please compare method which was used with method describe for the first time by Swain and Hillis in 1957. Use proper citation.

The citation of reference in text should be  done as is required by IJMS journal improved in almost all manuscript.

Native speaker should improve English in manuscript. 

Comments on the Quality of English Language

In opinion of reviewer the manuscipt shoudl be improved by the native speaker.

Round 2

Reviewer 2 Report

Comments and Suggestions for Authors

The authors improved manuscript. But still they following questions are addressed to the authors;

In lines 194-195 authors cited the article publish  in 2007. These only data concerning the intake of flavonoids. In 2022 the following  article was published. Crowe-White KM, Evans LW, Kuhnle GGC, Milenkovic D, Stote K, Wallace T, Handu D, Senkus KE. Flavan-3-ols and Cardiometabolic Health: First Ever Dietary Bioactive Guideline. Adv Nutr. 2022 Dec 22;13(6):2070-2083. doi: 10.1093/advances/nmac105

 Please compared your results with  recommendations from this article.

In line 195 please clarified g DM or fresh mass.

Line 198 the citation is wrong it should be Nadpal et al. [42]. This was not changed from previous version. Please make changes in all manuscript.

.

Line 212 please change „degenerative diseases…” non-communicable diseases.

Line 216 “Antioxidant ability means delay of oxidation process”  please correct this sentence because antioxidants usually scavenge free radicals and reduce oxidative stress.

Based on MDPI requirements all changes in manuscript should be highlighted or track changes function.

English should be improved by native speaker.

Comments on the Quality of English Language

Englsh should be imprved by native speaker
